# Validation of the Adapted eHEALS Questionnaire for Assessing Digital Health Literacy in Social Media: A Pilot Study

**DOI:** 10.3390/nursrep15090330

**Published:** 2025-09-09

**Authors:** Lucía Carton Erlandsson, Anna Bocchino, José Luis Palazón-Fernandez, Isabel Lepiani, Elena Chamorro Rebollo, Raúl Quintana Alonso

**Affiliations:** 1Salus Infirmorum Nursing and Physiotherapy Faculty, Pontifical University of Salamanca, 37002 Salamanca, Spain; lcartoner@upsa.es (L.C.E.); echamorrore@upsa.es (E.C.R.); rquintanaal@upsa.es (R.Q.A.); 2Fundación Hospitalarias Madrid, Calle del Doctor Esquerdo, 83, 28045 Madrid, Spain; 3Nursing Faculty “Salus Infirmorum”, University of Cádiz, Calle Ancha 29, 11001 Cadiz, Spain; anna.bocchino@ca.uca.es (A.B.); isabel.lepiani@ca.uca.es (I.L.)

**Keywords:** digital health, social networking, technology, health literacy, validation, scale

## Abstract

**Background:** Digital health literacy is crucial for navigating social media as a primary health information source. However, its interactive and unregulated nature fosters misinformation, requiring critical evaluation skills. Existing tools assess general internet use, but no validated instrument measures competencies specific to social media. This study aimed to adapt and validate the eHealth Literacy Scale (eHEALS) for this context. **Methods**: A content validation process was conducted with 33 experts, who evaluated the clarity, coherence, and relevance of the adapted questionnaire using item-level (I-CVI) and scale-level (S-CVI) content validity indices. A pilot study was then carried out with 250 participants to assess the instrument’s reliability and feasibility, measured through Cronbach’s alpha and McDonald’s Omega. **Results**: The adapted eHEALS demonstrated excellent content validity (S-CVI > 0.90) and high internal consistency (Cronbach’s α = 0.92; McDonald’s Ω = 0.92). The tool effectively captures key competencies for evaluating and applying health information in social media contexts, with exploratory factor analysis confirming a unidimensional structure that explained over 60% of the variance, supporting its robustness for use in population-based studies. **Conclusions**: This validated instrument provides a reliable method for assessing digital health literacy in social media, supporting the development of educational interventions to enhance critical appraisal skills and mitigate the impact of misinformation.

## 1. Introduction

In recent decades, patterns of information consumption in society have undergone a significant transformation, particularly in the domain of health-related information [1,2]. Social media has emerged as one of the primary sources of information for the general population, substantially displacing traditional media [1,3]. The increasing reliance on platforms such as Facebook, YouTube, and Instagram for health-related information has facilitated more immediate and dynamic access to healthcare content, enabling users to obtain information on diseases, treatments, and well-being in a manner that is both convenient and economical [4,5,6].

Social media has become a space where people seek health-related information, and this brings both positive aspects and significant challenges. Among the advantages are users’ access not only to a wealth of information, but also to the exchange of experiences and support from others facing similar health concerns [7,8,9]. These interactions serve as an additional form of health education, offering users practical suggestions and examples that help them feel more confident and capable of managing their own care [10]. Furthermore, when health professionals use these platforms to interact directly with users, communication can strengthen health literacy and encourage the adoption of healthier behaviors in everyday life [7].

However, social media also poses substantial limitations as a source of health information. The lack of stringent regulation and the rapid dissemination of content make it challenging to ensure the accuracy and quality of the information available. The ease of access to platforms where individuals without medical training can publish health-related content contributes to the proliferation of misinformation and pseudoscience, a phenomenon that became particularly evident during public health emergencies, such as the COVID-19 pandemic [11,12,13]. This “infodemic” on social media represents a tangible public health risk, generating confusion, anxiety, and even encouraging harmful health behaviors [14]. These challenges highlight the urgent need to develop tools that enable users to critically assess the reliability of the health information they encounter on social media.

Given this landscape, it is essential to assess users’ digital health literacy in the context of social media to promote the safe use of these platforms and improve the quality of the information they receive. Digital health literacy encompasses not only the ability to access and comprehend health information but also the capacity to critically evaluate and appropriately apply it in daily life [15]. Recent studies indicate that high levels of digital health literacy can protect users from misinformation and enhance their ability to manage their own health, particularly when critical thinking is encouraged and trust in reliable sources is strengthened. As a result, digital health literacy is increasingly recognized as a key determinant of health [16].

In this context, several instruments have been developed to measure digital health literacy in the broader internet environment. Among them, the eHealth Literacy Scale (eHEALS) [17]—originally designed to assess health literacy in relation to internet use—remains the most widely applied and has been adapted across diverse populations and digital settings. Other tools have also been introduced, such as the Digital Health Literacy Instrument (DHLI) developed by van der Vaart and Drossaert (2017) [18] and its Brazilian adaptation for adolescents (DHLI-BrA) validated by Barbosa et al. (2024) [19], which extend the assessment to a wider range of skills and cultural contexts. However, none of these questionnaires specifically addresses the use of social media for health-related purposes. A recent scoping review by Faux-Nightingale et al. (2022) [20] confirmed that, despite the growing relevance of social media as a primary source of health information, no validated instrument currently exists to evaluate digital health literacy in this setting. This gap underscores the need to adapt and validate measurement tools that can capture users’ competencies in navigating, appraising, and applying health information within social media environments.

Therefore, this study had two aims: (1) to adapt the eHealth Literacy Scale (eHEALS) to explicitly assess competencies for finding, appraising, and using health information on social media; and (2) to examine its content validity and preliminary reliability and factor structure in a pilot sample from the general population.

## 2. Material and Methods


**Phase 1. Adaptation and Translation Process**


Prior to the validation stages, the original English version of the eHEALS was adapted to explicitly assess health information in the context of social media. Each item was reformulated to replace references to “the Internet” with “social media platforms” (e.g., Facebook, Instagram, YouTube), while preserving the original construct. The questionnaire underwent a process of translation and cross-cultural adaptation following international recommendations [21]. First, two independent bilingual translators, native Spanish speakers with advanced English skills, performed a direct translation from the original version. Subsequently, a third translator, a native English speaker with no prior knowledge of the original version, performed the reverse translation.

The versions obtained were reviewed by the research team, made up of professionals in nursing, health communication, and psychometric methodology, selected based on their academic and practical experience in digital health literacy. The panel evaluated the semantic, idiomatic, and conceptual equivalence of each item, proposing adjustments when necessary until consensus was reached.

Finally, a cognitive adaptation was carried out through cognitive interviews with ten participants from the target population, with the aim of verifying the comprehensibility, relevance, and cultural appropriateness of the items. The observations collected were used to refine the final version prior to the content validation phase.


**Phase 2. Validation of the instrument’s content**


Content validity indicates whether the items of an instrument accurately represent the concept being measured. It is usually evaluated by a panel of experts who assess each item for clarity, coherence, and relevance.

Specifically, clarity refers to the comprehensibility of the items in terms of syntax and semantics; coherence assesses the logical alignment of the item with the overall scale; and relevance determines whether the inclusion of the item is necessary to ensure an adequate representation of the construct.


**Selection of the expert panel**


The questionnaire validation was conducted through the selection of an expert panel using purposive sampling, following predefined criteria. Experts (*n* = 16) were selected through purposive sampling, following the methodological recommendations of Polit and Beck (2004) [22]. Eligibility criteria included: (a) advanced academic or professional training in nursing, psychology, communication, or health sciences; (b) research or professional experience in health literacy, digital communication, or education; and (c) at least three years of professional experience. Candidates were identified through academic networks and professional associations and invited via institutional mailing lists. Only those who provided informed consent participated. In addition, a complementary panel of lay participants (*n* = 17) with high digital literacy was recruited to evaluate comprehension and cultural relevance. Lay participants were selected based on high digital literacy, assessed through a short screening questionnaire adapted from the Digital Health Literacy Instrument (DHLI). The screening evaluated frequency of social media use, diversity of platforms used, and self-reported ability to search, critically evaluate, and share health-related information online. Only individuals scoring above the 75th percentile were included in this group. Their participation contributed to assessing the clarity, comprehensibility, and cultural relevance of the adapted questionnaire from the perspective of the target audience.


**Validation session**


After the research team presented the study objectives, the expert panel assessed the clarity, coherence, and relevance of each questionnaire item using a survey administered via Google Forms. To ensure ethical compliance, all participants signed an informed consent form prior to their involvement.

During the validation session, each item of the questionnaire was evaluated independently for clarity, coherence, and relevance. These three dimensions were assessed separately using a 4-point Likert scale (1 = not at all clear/coherent/relevant, 4 = very clear/coherent/relevant). This procedure allowed the computation of item-level (I-CVI) and scale-level (S-CVI) Content Validity Indices for each dimension. In addition, an open-ended section was included to collect comments and suggestions for improving the items.


**Data analysis**


The statistical analysis involved calculating both the Item-Level Content Validity Index (I-CVI) and the Scale-Level Content Validity Index (S-CVI). The I-CVI reflected the proportion of experts who rated an item as valid, with values above 0.78 indicating excellent content validity [22].

To assess the overall validity of the instrument, the S-CVI was computed as the mean of all I-CVI values. An S-CVI greater than 0.90 was interpreted as evidence of excellent content validity for the entire questionnaire [22].


**Preliminary Version of the Questionnaire**


Following an in-depth analysis and the incorporation of recommendations obtained during the peer-review phase, the preliminary version of the questionnaire was developed. This version included both original items and revised elements, ensuring greater clarity, coherence, and representativeness of the construct. The preliminary questionnaire consisted of only eight items.


**Phase 3. Pilot study**


Following the integration of expert feedback, a pilot study (*n* = 250) was carried out between February and March 2024 using a non-probabilistic convenience sample.

Data were collected online through Google Forms, and the questionnaire was completed individually and anonymously, without the presence of a researcher. Recruitment was carried out via institutional mailing lists and social media platforms. Participation was voluntary and without incentives.

The objective of the pilot study was to validate the revised version of the instrument and assess key aspects such as question comprehension, language clarity, completion time, and any other relevant observations for further improvement, to assess the reliability of the instrument and its factor structure.


**Data analysis**


The data obtained from the pilot study were used to evaluate the instrument’s reliability through Cronbach’s alpha and McDonald’s Omega [23], which assess internal consistency. A preliminary analysis was also conducted to estimate the required sample size for future studies at the population level. This phase helped identify areas for improvement and reinforced the methodological soundness of the instrument for broader application.

To explore the underlying factor structure of the digital health literacy in social media questionnaire, an exploratory factor analysis (EFA) was performed using Principal Components as the extraction method. Prior to conducting the EFA, the adequacy of the data was assessed through Bartlett’s test of sphericity and the Kaiser–Meyer–Olkin (KMO) measure. The number of factors retained was determined using the Kaiser criterion (eigenvalues greater than 1) and by examining the inflection point of the scree plot [24]. All data analyses were performed with SPSS v29.

## 3. Results


**Phase 1. Adaptation and translation**


The forward–backward translation process confirmed the semantic equivalence between the Spanish version and the original English eHEALS. Minor wording adjustments were made during cognitive pre-testing (*n* = 10) to improve readability and contextual clarity, particularly replacing general references to “the Internet” with “social media platforms.” All items were considered comprehensible and relevant by participants, and no item was excluded at this stage. The final adapted Spanish version was then submitted to expert validation.


**Phase 2. Validation of the instrument’s content**


The validation sample comprised two complementary groups. The first group consisted of 16 experts with professional profiles in health, teaching, and communication, including university professors, nurses, midwives, psychologists, and health content creators. Of these participants, 56.3% were women and 43.7% men, with a mean age of 31.4 years. Of the 16 experts, nine (56.3%) held doctoral degrees and seven (43.7%) were university professors. Their fields of expertise were nursing (*n* = 5), psychology (*n* = 3), communication (*n* = 2), and health sciences education (*n* = 6). This profile ensured that the panel provided both academic and professional expertise relevant to digital health literacy and instrument validation.

The second group included 17 lay participants from the general population with high digital literacy and frequent social media use. In this group, 52.9% were female and 47.1% male, with a mean age of 26 years. Regarding educational level, 70.6% had university studies, 17.6% vocational training, and 11.8% secondary education.

The results of the validation process for the adaptation of the eHEALS questionnaire for use in social media are presented in Table 1, which displays the item-level content validity index (I-CVI) for clarity, coherence, and relevance. Across all three evaluated aspects (clarity, coherence, and relevance), all questionnaire items achieved I-CVI values exceeding 0.78, indicating excellent content validity according to established criteria. The calculated scale-level content validity index (S-CVI) for each dimension was 0.93 for clarity, 0.95 for coherence, and 0.98 for relevance, reinforcing the overall quality of the instrument in terms of item representativeness and appropriateness. These findings demonstrate that the questionnaire meets high standards of excellence in its design and content.

The items that received the highest scores in terms of relevance were items 1 and 4–8. Items 3–7 achieved the highest scores for clarity, while items 4–7 obtained the maximum score for coherence. However, it is important to note that none of these items were excluded, as their I-CVI values were deemed acceptable.

In the panel composed from the general population, the results were even more favourable (Table 2), with both the I-CVI and S-CVI reaching the maximum value of 1 across all dimensions—clarity, coherence, and relevance.


**Phase 3. Pilot study**


The questionnaire was administered to a pilot sample of 250 individuals from the general population (Table 3) to assess the clarity, applicability, and feasibility of the instrument, as well as to identify any potential difficulties in its administration. This phase aimed to ensure that respondents fully understood the questions and to determine the time required to complete the questionnaire. Based on the results obtained from the pilot sample, necessary modifications were made to the final version of the questionnaire, including adjustments to the wording and the order of certain questions.

Both the Cronbach’s alpha and McDonald’s omega showed high internal consistency (α = 0.92; Ω = 0.92). A preliminary estimate indicated that 69.6% of participants used social media for health information, which was used to calculate the required sample size for a larger population-based study (*n* = 913).


**Exploratory factor analysis (EFA)**


The KMO value was 0.884, which is considered meritorious [24] and indicates a high proportion of common variance among the items. The χ^2^ value of Bartlett’s sphericity test was statistically significant (*χ*^2^_28_ = 1475.78, *p* < 0.001), which indicated that the results were suitable for exploratory factor analysis. The scree plot revealed the presence of a single factor (Figure 1) with an eigenvalue of 5.11 which accounted for 63.87% of the total variance, which is over the threshold of 60% to consider the model adequate [25]. The communalities of the items were all above 0.50 (0.514–0.772) suggesting that the items explain a substantial proportion of their variance through the common factor. All item loadings were >0.70 (Table 4), so the 8 items were retained.

## 4. Discussion

Access to health information has undergone significant transformations in recent decades, not only altering the sources used by the population but also reshaping how this information is interpreted and applied in decision-making [26]. The continuous increase in social media use has contributed to the emergence of a new landscape, enabling a constant flow of digital content—often without users actively seeking it [27]. This shift presents unique challenges regarding digital health literacy, a field that has been extensively studied in general internet contexts but remains insufficiently explored within the specific domain of social media. The present study addresses this gap by providing an adapted and validated tool to assess critical competencies in a dynamic and frequently misinformation-prone environment.

Norman and Skinner [17], in their initial validation of the eHEALS, demonstrated that this instrument is a useful tool for measuring how users perceive their ability to search for, comprehend, and apply online health information. However, its design does not account for interaction, virality, or the social dynamics inherent to social media—factors that justified the need for this study.

Our results indicate that the adaptation and validation of the eHEALS maintain its optimal internal consistency (Cronbach’s α = 0.92: McDonalds Ω = 0.92), aligning with values reported in the instrument’s original validation [17]. At the same time, as highlighted by previous research, digital health literacy is a key determinant in improving health behaviors and reducing vulnerability to misinformation [28]. These findings are comparable to studies that have validated similar instruments, such as the Digital Health Literacy Instrument (DHLI-BrA) used by Barbosa [19] to measure digital health competencies among adolescents in internet-based settings. However, the adapted eHEALS has the distinct advantage of focusing specifically on social media, addressing a gap identified by other authors [29], who emphasized the need to develop specialized tools for interactive environments, recognizing the increasing influence of social media and other digital platforms in shaping how young individuals access and utilize health information.

The content validity analysis, supported by the I-CVI and S-CVI indices, confirms that experts deemed the questionnaire items to be relevant, clear, and coherent in assessing digital health literacy within social media contexts. This suggests that the semantic adaptation has not compromised the instrument’s ability to accurately measure this construct.

Recent studies have identified that digital health literacy in social media is not solely dependent on information-seeking abilities but also on the capacity to interpret, filter, and share content within an interactive ecosystem—where social validation and message virality can distort perceptions of informational accuracy [30]. In this regard, adapting the eHEALS questionnaire to the social media context represents a significant methodological advancement, as it enables a more precise evaluation of users’ ability to distinguish reliable sources in highly dynamic and misinformation-prone environments.

The adaptation and validation of the eHEALS, therefore, represent an important step in identifying factors that may predispose individuals to uncritical consumption of health-related information—an insight that could be pivotal for designing evidence-based educational interventions. Additionally, this validation provides key data on the digital competencies required to assess the quality of health information shared on social media and its impact on health decision-making. Finally, the questionnaire’s validation reinforces the idea that direct equivalencies cannot be assumed between digital health literacy in general internet use and in social media, as interaction dynamics, information overload, and content validation mechanisms function differently in each environment [31,32,33].

Finally, the exploratory factor analysis showed that the adapted scale maintains a unidimensional structure. This factorial solution is aligned with the theoretical structure of the original eHEALS scale, which also identified a single factor, thus favoring its applicability and comparability in different contexts. This result also indicates that the items assessed cluster coherently around a single construct: digital health literacy in the specific context of social networks. The strong correlation between the items suggests that they all contribute significantly to measuring this competency, which is particularly relevant to meet the study’s objective of having a useful and specific tool to assess such skills in today’s digital environment. The high internal consistency among the items, reflected in their high factor loadings, indicates that they all contribute significantly to the measurement of the same concept.

This study contributes to an emerging field of research by offering a validated tool that enables a more precise evaluation of digital health literacy in social media, facilitating the development of strategies aimed at enhancing the population’s critical capacity to counteract health-related misinformation.

This study has several limitations. The use of expert judgment introduces subjectivity, and the predominance of young and highly educated participants may limit the generalizability of the findings. Additionally, as the questionnaire was developed within a single cultural context, cross-cultural adaptations will be necessary for international use. Furthermore, since the instrument is based on self-report and administered through a convenience sample, participants’ responses may reflect subjective perceptions rather than actual digital health literacy skills. These aspects should be considered when interpreting the findings, and future studies should aim to include more representative samples and combine self-reported measures with objective performance-based assessments.

Finally, although the exploratory factor analysis supported a unidimensional structure consistent with the original version of the instrument, this unidimensionality could limit the questionnaire’s ability to capture more nuanced aspects of digital health literacy in social networks, such as critical thinking, interactive skills or trust in information sources. Future research could explore the presence of subdimensions through confirmatory factor analysis or other measurement models.

Although our design included an exploratory factor analysis (EFA), we acknowledge that a confirmatory factor analysis (CFA) on an independent sample would provide stronger evidence for the structure of the instrument. We have noted this as a limitation and suggested that future studies should incorporate CFA or, at minimum, parallel analysis to enhance robustness.

Finally, beyond these psychometric considerations, it would be important for future studies to explore the role of artificial intelligence (AI) in shaping the health information available on social media. Future research should consider how to integrate AI into health literacy frameworks, given its rapid evolution and influence on social platforms.

## 5. Conclusions

Our results suggest that the adaptation and validation of the eHEALS questionnaire for the context of social media constitutes a reliable and valid instrument for measuring digital health literacy, facilitating the identification of training needs and the design of educational interventions aimed at the safe and effective use of these platforms.

The implications of this study on digital health literacy are numerous and significant, particularly in the design of educational interventions. The findings can guide the development of educational programs and resources aimed at enhancing digital health literacy skills, especially among populations with low educational attainment and limited access to technology. The tool developed in this study provides a foundation for identifying specific digital competencies that should be reinforced in educational campaigns.

Moreover, addressing disparities in digital health literacy is crucial to ensuring equitable and effective access to digital health tools. Enhancing digital literacy can have a positive impact on health outcomes and quality of life by facilitating the comprehension and use of digital health resources, improving communication with healthcare professionals, and promoting healthy behaviors.

## Figures and Tables

**Figure 1 nursrep-15-00330-f001:**
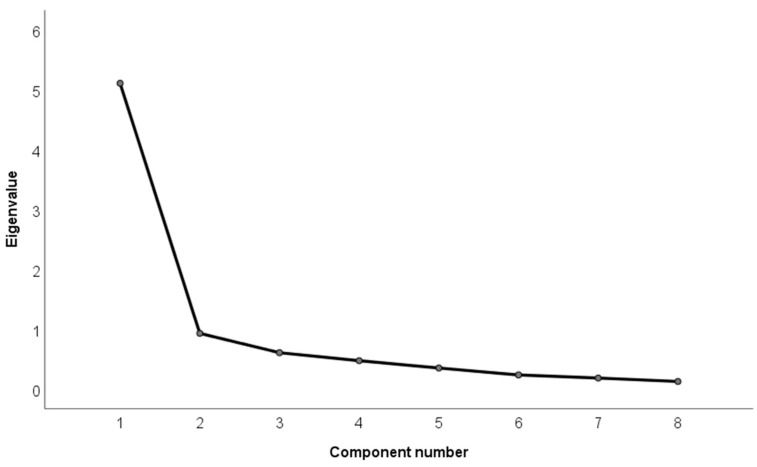
Scree plot of the factors of the eHEALS questionnaire for use in social media extracted by principal component analysis.

**Table 1 nursrep-15-00330-t001:** Content Validity Indexes (I-CVI and S-CVI)–Specialists.

Item	Clarity		Coherence		Relevance	
	I-CVI	Expert Agreeing	I-CVI	Expert Agreeing	I-CVI	Expert Agreeing
- Conozco qué recursos sobre salud están disponibles en redes sociales	0.81	13/16	0.88	14/16	1.00	16/16
2. - Sé dónde puedo encontrar recursos útiles sobre salud en redes sociales	0.88	14/16	0.94	15/16	0.94	15/16
3. - Sé cómo puedo encontrar recursos útiles sobre salud en redes sociales	1.00	16/16	0.94	15/16	0.94	15/16
4. - Sé cómo utilizar las redes sociales para encontrar respuestas a mis cuestiones de salud	1.00	16/16	1.00	16/16	1.00	16/16
5. - Sé cómo utilizar la información sobre salud que encuentro en redes sociales para que me ayude	1.00	16/16	1.00	16/16	1.00	16/16
6. - Tengo las habilidades necesarias para evaluar los recursos sobre salud que encuentro en redes sociales	1.00	16/16	1.00	16/16	1.00	16/16
7. - Puedo distinguir los recursos de salud de alta calidad de los recursos de salud de baja calidad que se encuentran en redes sociales	1.00	16/16	1.00	16/16	1.00	16/16
8. - Tengo confianza a la hora de utilizar la información de redes sociales para tomar decisiones sobre salud	0.75	12/16	0.88	14/16	1.00	15/16
S-CVI *	0.93		0.95		0.98	

* S-CVI = average of the I-CVIs.

**Table 2 nursrep-15-00330-t002:** Content Validity Indexes (I-CVI and S-CVI) in the General Population.

Item	Clarity		Coherence		Relevance	
	I-CVI	Expert Agreeing	I-CVI	Expert Agreeing	I-CVI	Expert Agreeing
Item 1	1.00	16/16	1.00	16/16	1.00	16/16
Item 2	1.00	16/16	1.00	16/16	1.00	16/16
Item 3	1.00	16/16	1.00	16/16	1.00	16/16
Item 4	1.00	16/16	1.00	16/16	1.00	16/16
Item 5	1.00	16/16	1.00	16/16	1.00	16/16
Item 6	1.00	16/16	1.00	16/16	1.00	16/16
Item 7	1.00	16/16	1.00	16/16	1.00	16/16
Item 8	1.00	16/16	1.00	16/16	1.00	16/16
S-CVI *	1.00		1.00		1.00	

* S-CVI = average of the I-CVIs.

**Table 3 nursrep-15-00330-t003:** Sociodemographic characteristics of the pilot sample.

Variable	*n* = 250
Gender	data
Female	48.8
Male	50.0
Other	1.2
Age	35.2 ± 15.4
Educational level	
No studies	1.2
Primary studies	0.4
Secondary studies	16.0
Vocational training	20.0
University studies	55.2
Other	7.2

Quantitative variable (age) is expressed as median (IQR). Qualitative variables are expressed as frequency (n) and percentage (%).

**Table 4 nursrep-15-00330-t004:** EFA Factor Loadings of Digital Health Literacy in Social Media Items.

Items	Mean (sd)	Loadings	Communalities
1	2.57 (1.18)	0.732	0.536
2	2.65 (1.15)	0.857	0.735
3	2.73 (1.19)	0.879	0.772
4	2.76 (1.16)	0.853	0.728
5	2.87 (1.17)	0.844	0.713
6	2.97 (1.27)	0.765	0.586
7	3.10 (1.25)	0.725	0.525
8	2.57 (1.22)	0.717	0.514

## Data Availability

The data presented in this study are available on request from the corresponding author. The data are not publicly available due to privacy.

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
