# Peer review of "Validation of the Adapted eHEALS Questionnaire for Assessing Digital Health Literacy in Social Media: A Pilot Study"

_nursrep, 2025, doi:10.3390/nursrep15090330_

Round 1

Reviewer 1 Report

Comments and Suggestions for Authors

This study focuses on the adaptation and validation of a digital health literacy measurement instrument for use on social media platforms. This is a relevant and current topic, and the instrument has significant potential to contribute to the literature on health literacy and to be used to inform potential public health policies.

However, there are some points in the work that need clarification, especially regarding the method. I list them below:

  1. In the introduction, some information is presented without citing the source. For example, in the paragraph beginning on line 66, eHEALS is introduced without citing the original article, which is cited only in the following paragraph. In the paragraph beginning on line 70, it is stated that "Several validated questionnaires exist to measure digital health literacy in the broader internet context." In this case, it would be important to cite some of these studies. Later, it is stated that "other validated instruments have been developed..." but again without citing the instruments.
  2. The objective of this study is to adapt and validate eHEALS. However, the work is limited to describing the validation process, omitting how the adaptation was performed.
  3. I was unable to identify some important information in the research process. Some questions remain unanswered: Where was the instrument administered? In what language? Was it translated? How was this translation done? How was it adapted for social media?
  4. Regarding the validation process, some questions remain. First, how were the expert panel members selected?
  5. In line 103, it is stated that the panel members are "experts." However, in the paragraph beginning on line 150, there are indications that the panel members are not experts on the topic. This needs to be clarified. Who are these "experts"? What are they "experts" in? How many of them are experts in health literacy or digital literacy?
  6. Line 109 states that the expert panel's data collection instrument contained questions assessing clarity, coherence, and relevance. It is unclear whether these aspects were assessed together or separately.
  7. Regarding the pilot study, the authors state in line 128 that there were 250 respondents, but only present the characteristics of these respondents in line 198. It's important to present the sample profile. Another important point is to detail how the questionnaires were administered? Was the application done in the presence of the researcher? When and where were the questionnaires administered?
  8. Regarding data analysis, there is a lack of information on data processing and distribution.
  9. In line 138, and elsewhere in the text, the authors highlight the use of Cronbach's alpha to assess the instrument's reliability. However, it is important to verify the adequacy of this measure, which has problems related to its assumptions. The use of McDonald's omega tends to be a more robust measure. It would also be interesting to evaluate the omega without the item. On this topic, see the works of: Viladrich et al., 2017 and Revelle and Zinbarg, 2009.
  10. In line 143, the authors state that "an exploratory factor analysis (EFA) was performed using principal component analysis (PCA)." I don't understand this statement. I understand that EFA and PCA are distinct analysis methods. Principal component analysis and exploratory factor analysis use item variance to reduce variance to components or factors. However, PCA is based solely on the linear correlation of the observed variables, not differentiating between the common variance and the specific variance of each variable. EFA, on the other hand, considers only the variance shared by the items. Thus, in PCA, items tend to have higher factor loadings and commonalities than in EFA. In other words, this needs to be clarified. Furthermore, it is important to indicate which estimator was used in the analysis, as well as the software and statistical packages.
  11. In line 154, when presenting the profile of the expert panel, it states that "many held doctoral degrees or teaching positions, providing key expertise for the content validity analysis." I emphasize that the scientific language must be precise; that is, how many respondents hold a doctorate degree? How many are university professors? What is their field of expertise?
  12. In line 156, how was it measured that the 17 participants (some of whom had low educational levels) had “high digital literacy”?
  13. I believe it's important to add a discussion about generative AI. The use of this type of tool has great potential to spread misinformation on social media.

Author Response

Dear reviewer, we sincerely appreciate your valuable comments and suggestions. Please find attached our detailed response to each of them

Reviewer 2 Report

Comments and Suggestions for Authors

The authors should include the adaptation they made to the questionnaire, as this was their main objective. The objective of this study is to adapt and validate the eHEALS questionnaire to specifically measure digital health literacy in the use of social media, addressing a critical gap in the evaluation of these competencies and facilitating the design of educational interventions for the safe and effective use of these platforms. They should also state whether they made any modifications to the instrument and specify what these were. The authors should not only include the number of the item but also provide the full content of each item in the main text of the Results and Discussion sections. These elements allow for a clearer understanding of the information being presented, described, and discussed. In the Conclusions section, the authors should include a sentence that clearly and specifically summarises the study, drawing directly on the data generated by their own research.

Author Response

(The authors gave the same response as above.)

Reviewer 3 Report

Comments and Suggestions for Authors

Dear authors,

The study is undoubtedly truly necessary. It poses how information is currently being received through social media. However, the method is insufficient and unclear. The items being evaluated are not known and it is not clear what they really want to assess. The different criteria presented are vague, lacking detail, and could apply to any situation. Therefore, the rest of the article loses interest by not knowing what they are really trying to analyze. 

Best regards.

Author Response

(The authors gave the same response as above.)

Reviewer 4 Report

Comments and Suggestions for Authors

The paper is well-structured, methodologically sound, and makes a valuable contribution by adapting and validating eHEALS for the social media context. The study is clearly presented, with strong psychometric results and relevant implications for practice.

I would only recommend minor revisions:

  1. Please expand the discussion of limitations, particularly regarding the use of a convenience sample and the self-perceived nature of the measure.

  2. Consider briefly elaborating on how the unidimensional structure may limit the capture of more nuanced aspects of digital health literacy (e.g., critical thinking, trust in sources).

With these clarifications, the manuscript will be further strengthened.

Author Response

(The authors gave the same response as above.)

Reviewer 5 Report

Comments and Suggestions for Authors

Abstract: The background and method are specifically presented in the abstract, but the results are only too brief. Please present the results more specifically.

introduction 

No analysis of the results of previous studies related to eheals has been presented. Please analyze various previous studies using eheals and clearly present the purpose of how you intend to re-verify the questionnaire in this study by analyzing how it is being used. If there is no previous study related to eheals, please supplement it with the content that the purpose is to validate ehealth through analysis of previous studies using similar questionnaires.

Method of research

Was it conducted without any specific criteria for selecting experts? Please provide specific criteria for 33 expert panels.

the results of the study 
I think the study result 1 went well properly
In Study Results 2, the values related to the efa analysis results should be presented in the table, but the table simply presents the average. Please present all the results presented in the second stage pilot study in the table.

In the discussion, we are discussing well with previous studies related to eheals. However, it would be nice to add information on the strengths and weaknesses of this questionnaire compared to previous studies of eheals and similar questionnaires

a general opinion 

Currently, the plagiarism rate on the posting system is 26%. Please review the overall sentence structure. Other than that, the overall content is well organized. Please review it carefully to increase the completeness.

Author Response

(The authors gave the same response as above.)

Reviewer 6 Report

Comments and Suggestions for Authors

This study addresses a timely and important gap by adapting and validating the eHEALS questionnaire for assessing digital health literacy specifically in social media contexts. The rationale is clearly stated, and the two-phase design (content validation + pilot) is logical. The statistical results (S-CVI > 0.90, Cronbach’s α = 0.92, unidimensional factor structure) are strong and support the scale’s reliability.

However, several aspects require improvement before the manuscript can be considered for publication:

  • Although well-organised, it exhibits redundancy (e.g., repeating advantages/disadvantages of social media) and should incorporate more contemporary literature on digital health literacy and validation studies.
  • Provide more detail on the linguistic adaptation process, particularly forward–back translation and expert consensus steps, to improve reproducibility for international contexts.
  • The pilot sample is skewed toward younger, highly educated participants; the impact of this on generalizability should be discussed in more depth.
  • Only exploratory factor analysis was performed. A confirmatory factor analysis (CFA) on a separate sample, or at a minimum parallel analysis, would strengthen evidence for unidimensionality.
  • Very high inter-item correlations suggest possible redundancy. Reporting item–total correlations and considering whether a shorter form could maintain psychometric strength would be valuable.
  • Expand to acknowledge the self-report nature of the instrument, cultural specificity, and potential limitations in capturing nuanced competencies such as critical thinking or interactive skills.
  • Tables and figures are precise, but captions should be more descriptive, summarising the main takeaways. Minor English language refinements are recommended to improve clarity and flow.

With these revisions, the manuscript would present a more robust and internationally relevant validation study that could have significant utility in public health, education, and digital literacy interventions.

Comments on the Quality of English Language

The English is clear overall but would benefit from minor editing to remove redundancies and improve flow.

Author Response

(The authors gave the same response as above.)

Round 2

Reviewer 1 Report

Comments and Suggestions for Authors

Thank you for the opportunity to revisit this manuscript, which has improved greatly since the last version. The content is scientifically sound and the work is well presented. I believe there is great potential for academic contribution and impact. 

Author Response

We sincerely thank you for your positive and encouraging feedback. We are pleased to hear that the improvements made to the manuscript were well received and that you consider the work scientifically sound and impactful. We greatly appreciate your constructive comments throughout the review process, which have undoubtedly strengthened our study.

Reviewer 3 Report

Comments and Suggestions for Authors

Dear authors

Thank you for taking into account the recommendations. I believe the clarifications have helped improve the article.

Author Response

We sincerely thank you for your thoughtful review and for recognizing the clarifications made. Your recommendations have been very helpful in improving the quality and clarity of the article.

Reviewer 6 Report

Comments and Suggestions for Authors

Thank you for your thorough revisions. The manuscript is now clearer, more concise, and supported with updated references. The adaptation and translation process is well-detailed, the limitations are explicitly acknowledged, and tables/figures are improved with clearer captions. The English language is also smoother.
My only minor suggestion is to consider adding an English translation of the Spanish items (in a column or appendix) to enhance accessibility for international readers. This is optional.
Overall, the study is well-prepared for publication and makes a valuable contribution to digital health literacy in social media contexts.

Author Response

We sincerely thank you for your constructive and encouraging feedback. We are pleased that the revisions improved the clarity, conciseness, and overall quality of the manuscript. Regarding your kind suggestion to include an English translation of the Spanish items, we would like to clarify that the items were presented in Spanish because this was the language in which the validation process of the scale was conducted. For this reason, and to maintain methodological consistency, we have kept them in their original language.

We greatly appreciate your thoughtful recommendation and your recognition of the contribution of this study, which further motivates us in our work.